# Occupational Exposure to Diesel Particulate Matter in Western Australian Mining: A Retrospective Analysis and Challenges to Future Compliance

**DOI:** 10.3390/ijerph22091412

**Published:** 2025-09-10

**Authors:** Matthew Oosthuizen, Kerry Staples, Adelle Liebenberg, Kiam Padamsey, Marcus Cattani, Andy McCarthy, Jacques Oosthuizen

**Affiliations:** 1School of Medical and Health Sciences, Edith Cowan University, Joondalup Campus, Joondalup, WA 6027, Australia; m.oosthuizen@ecu.edu.au (M.O.); a.liebenberg@ecu.edu.au (A.L.); k.padamsey@ecu.edu.au (K.P.); m.cattani@ecu.edu.au (M.C.);; 2Department of Health (Western Australia), Epidemiology Directorate, Perth, WA 6004, Australia

**Keywords:** mining industry, occupational hygiene, respirable elemental carbon, underground mining, workplace exposure limit, workplace exposure standard

## Abstract

De-identified diesel particulate matter (DPM) exposure data (*n* = 24,459) was obtained from the Western Australian mining regulator to assess compliance with the current Workplace Exposure Standard (WES) of 0.1 mg/m^3^, measured as submicron elemental carbon, and a proposed limit of 0.01 mg/m^3^, assessed as respirable elemental carbon. R and R-Studio were used to generate summary statistics comparing compliance to the current and proposed limits, stratified by industry and occupational groups. To examine temporal trends, a zero-adjusted gamma model was used to assess whether expected sample means changed over the past ten years, using commodity and location as covariates. DPM exposures have declined significantly over the past decade, and modelling indicates compliance with the current WES. However, the proposed limit introduces both a lower limit and a different sampling method, which present challenges. The sector most affected by these changes is underground gold mining. Several occupational groups, such as ground/roof support, shotfirer, long hole drill and blast, and production and services, are at highest risk of non-compliance. Meeting future exposure limits will require enhanced control strategies, including, cleaner fuels, reduction or elimination of diesel-powered machinery in underground operations and appropriate and regulated use of respiratory protective equipment when assessing compliance.

## 1. Introduction

Diesel particulate matter (DPM), a constituent of diesel exhaust emissions, is a waste product associated with incomplete combustion of diesel and is a recognised occupational hazard in mining. It consists of a complex mixture of fine particles that can be inhaled deep into the lungs. DPM typically consists of elemental carbon (EC) onto which organic carbon compounds and other particles, such as unburned fuel and metallic additives, are adsorbed. The majority of diesel particle mass falls within the fine particle size range (≤2.5 µm), enabling them to deposit in the alveolar region, thus increasing potential health risks [1].

Diesel-powered engines are widely used in the mining industry, both in open-cut and underground operations. Older vehicles and diesel engines that are not well serviced and maintained produce more pollutants such as oxides of nitrogen, carbon monoxide, hydrocarbons, and particulate matter [2]. The mining industry has seen improvements in DPM exposures over the last decade due to improvements in the composition and quality of diesel fuel and technological advancements in engines, including catalytic converters and filters. Substitution of diesel-powered plants with liquid petroleum gas (LPG), and, more recently, battery-operated equipment is also leading to significantly lowered DPM levels in mining, particularly when deployed in underground environments [3]. Despite technological advancements, underground miners still experience significantly higher exposure levels compared to surface miners, highlighting ongoing challenges in exposure management in underground environments [1].

Diesel engine exhaust is classified as a Group 1 carcinogen by the International Agency for Research on Cancer (IARC), underscoring the need for stringent workplace exposure controls to safeguard worker health [4]. Short-term health effects from diesel engine exhaust exposure notably include throat and bronchial irritation, headaches, nausea, cough, phlegm production, and wheezing, while chronic exposure significantly increases risks for respiratory and cardiovascular diseases [1]. There is also a risk of chemical asphyxiation due to carbon monoxide exposure, particularly in underground mines [5,6]. Epidemiological studies have consistently shown associations between occupational DPM exposure and adverse chronic health outcomes, particularly respiratory and cardiovascular diseases, and increased cancer risk [7,8,9,10]. Furthermore, specific occupational groups such as shotcrete operators, drill operators, and blast operators have been found to be particularly vulnerable to higher levels of DPM exposure and consequent health risks, necessitating targeted interventions to reduce exposures for these groups [11].

Currently, DPM exposures in the Australian mining industry are regulated using a workplace exposure standard (WES) of 0.1 mg/m^3^, measured as submicron elemental carbon (SMEC). The new proposed workplace exposure limit (WEL) of 0.01 mg/m^3^ based on respirable elemental carbon (REC), ref. [12], is lower than the revised European Union standard of 0.05 mg/m^3^ (elemental carbon), which was introduced for general industry in 2023, and from 21 February 2026, it will also apply in underground mining and tunnelling [13]. While in 2023, Safe Work Australia [14] had initially indicated that the revised exposure standard for DPM was likely to be set at 0.015 mg/m^3^, the final decision in 2025 established a more stringent limit of 0.01 mg/m^3^ [12]. This represents a 10-fold reduction in the numerical exposure limit and a shift in the particle size measured, complicating comparisons between historical and future exposure data and posing challenges for regulatory transition and industry compliance. While direct comparison between SMEC and REC values is technically limited, historical compliance with the proposed 0.01 mg/m^3^ WEL, even when based on SMEC, may conservatively indicate likely compliance with the respirable limit. This is yet an untested hypothesis that could be investigated further when REC sampling has been implemented. Although REC is part of the broader size range measured by SMEC, it is not a direct component in the sense that it is not a subset that can be easily isolated from the SMEC measurement. The two methods utilise different sampling and analytical techniques to target different particle size fractions, and it is important to note that compliance with one does not automatically ensure compliance with the other.

Future exposure assessments must be conducted using REC sampling methods to ensure alignment with the updated standard. This is expected to have wide-ranging impacts across the mining industry, particularly in the confined spaces of underground mines where diesel-powered plants and vehicles are still very prevalent. Recent studies emphasise the necessity for stringent occupational standards to mitigate health risks, highlighting that even the currently recommended exposure levels may still pose significant risks for chronic respiratory diseases and cancer, advocating for continuous improvement in monitoring and control strategies [1,11].

This research aims to determine the extent to which historical DPM exposure data from the Western Australian (WA) mining sector indicate likely compliance with the proposed REC-based WEL and to examine how temporal trends and operational contexts, such as underground versus surface mining, influence compliance. The study also considers the implications of the shift from SMEC to REC-based sampling and identifies SEGs at the highest risk of non-compliance under the proposed standard, and these SEGs are to be monitored proactively to determine compliance and identify where more controls may be required.

## 2. Methods

A senior authorised officer from the Department of Energy, Mining, Industry Regulation and Safety (DEMIRS), Perth, Western Australia, extracted all DPM personal exposure sampling data from their Safety Regulatory System (SRS), which was checked, verified, and de-identified before being provided to the research team in the form of an MS Excel spreadsheet. The data were stored in a secure, password-protected online system, in accordance with the Edith Cowan University (ECU) Data Management plan associated with ethics approval 2023-04914-Oosthuizen. It should be noted that the data were shift-adjusted using the pharmacokinetic shift adjustment model to accommodate the work rosters common to WA mining, generally 12 h shifts for multiple consecutive days.

All DPM exposure results in the dataset were reported as SMEC (mg/m^3^), consistent with the current WES. Subsequently, data were analysed using the statistical software R and R Studio (R Core Team, 2024). Summary statistics were produced by commodity type and occupational group. Results were compared to current and proposed exposure standards. Occupation codes, location codes and associated DPM results were reviewed to identify the Similar Exposure Groups (SEGs) impacted by these changes and evaluate the anticipated impact of the proposed WELs. Descriptive statistics were generated from the cleaned dataset, including the mean, lower 95% confidence limit (LCL_95%_), and upper 95% confidence limit (UCL_95%_) to assess compliance for various SEGs. The dose–response curve for DPM is linear [15], so the arithmetic mean was selected, as it better represents the average population risk than a geometric mean, which may underestimate the risk [16].

The data are right-skewed, continuous, and positive, and follow a gamma distribution. To examine recent temporal trends, a zero-adjusted gamma model (ZAG) was used to assess whether the expected sample means changed over the past ten years, using commodity and location as covariates. The gamma distribution is strictly positive and does not contain zero. To account for samples below the DPM limit of detection of 0.01 mg/m^3^, which are recorded as zero, a zero-adjusted gamma model was required. The gamma model within the generalised linear model has been shown to be more accommodating of right-skewed samples than the lognormal distribution [17].

For the purpose of this study, the term diesel particulate matter (DPM) workplace exposure limit (WEL) refers specifically to respirable elemental carbon (REC), which underpins the proposed limit of 0.01 mg/m^3^ REC. However, historical DPM exposure data from the WA mining sector were collected using the NIOSH5040 respirable sampling method [18], and results were reported as SMEC (mg/m^3^), in accordance with current mining regulations in WA, New South Wales, and Queensland. Since SMEC and REC sampling methods capture different particle size fractions (REC versus SMEC), direct comparisons between historical SMEC data and the proposed REC-based WEL are problematic. This methodological discrepancy must be considered when interpreting historical compliance trends, as it limits the precision with which historical data can be used to assess future compliance.

## 3. Limitations

The dataset exhibits inherent variability due to differences in sampling conditions, operational practices, and reporting accuracy.

The bulk upload to the SRS database relies on occupational hygienists to accurately record worker shift patterns and to consistently make the correct shift adjustment.

In the absence of co-located REC measurements or validated conversion factors between SMEC and REC, direct comparisons should be interpreted with caution. It is assumed that all historical samples were collected using the SMEC method. While this allows for confident comparisons with the current WES (based on SMEC), comparisons to the proposed WEL (based on REC) are likely to be inaccurate.

Recorded SMEC values are expected to overestimate REC concentrations, as REC represents only a fraction of SMEC.

It should also be noted that the uncertainty range of the NIOSH5040 method is +/−16.7% at 23 ug/m^3^ [18].

## 4. Results

A total of 24,459 DPM samples were received from DEMIRS, and all were included in the subsequent analysis. To account for variations in sampling time, adjusted sample results were calculated using the adjusted exposure standard provided in the dataset. No other data cleaning was required. The range of the sample data was 0–5.5 mg/m^3^. No data points were considered outliers. A total of 163 samples (0.67%) were recorded as zero. Zeros were randomly distributed across location and occupation. The lack of extreme values and the small proportion of zero sample results mean the ZAG model is unlikely to be affected by either. As shown in Table 1, the majority of samples (n = 17,445; 71.3%) were collected from underground operations. Most of these underground samples originated from gold mining operations, which represent a key area of concern for airborne diesel particulate emissions.

As previously mentioned, SMEG and REC results cannot be directly compared, and so these results need to be considered as indicative of probable compliance. The arithmetic means and 95% confidence limits (CLs) for DPM samples from 2014 to 2024 (Figure 1) show a significant reduction in exposures over this period, with the mean and UCL_95%_ well below the current WES of 0.1 mg/m^3^ SMEC over the entire timeframe, and it can be seen that although results are trending closer towards compliance with the proposed WEL of 0.01 mg/m^3^ REC, mean exposures and CLs are still above the proposed WEL, particularly for underground mining. Visual inspection of the results shows that the mean for underground locations is consistently higher than the expected mean for surface locations, and this was confirmed by the ZAG model analysis (*p* < 0.0001).

The sample means by commodity type are shown in Figure 2. The means for all commodity types have been significantly below the current WES for the past 10 years. The means for copper, lead, zinc, gold, and nickel sectors have declined rapidly since 2014 and by 2024 were all clustered around the proposed WEL, with only gold and nickel having mean values above the new WEL.

Figure 3 expands on the data presented in Figure 2 by disaggregating mean DPM exposures by commodity and including 95% CLs. If a commodity type has 95% upper and lower CLs above the proposed WEL (blue line), it suggests the sample mean is significantly above the proposed WEL. If the 95% CLs are both below the line, it suggests the sample mean is significantly lower than the proposed WEL. Where the CL range crosses the proposed WES, it suggests there is no significant difference between the sample mean and the proposed WES. When comparing the 2024 exposure results to the proposed WEL of 0.01 mg/m^3^ (blue line), only the bauxite and iron ore sectors report mean concentrations and CLs below the proposed WEL. In contrast, exposures in copper, lead and zinc, gold, nickel, and the “other” commodity group are equal to or exceed the proposed WEL. For the diamond sector, no sample data were available beyond 2020.

To eliminate the influence of historically higher levels, an analysis to test compliance was performed using 2023 and 2024 data only, stratified by SEG and location (surface or underground). Occupations with a total sample size above 25 and UCL_95%_ above the future WEL are presented in Table 2 and Figure 4. The four occupational groups were all working in underground locations. Several geometric standard deviation (GSD) results are near or exceed the value of 3, which indicates high variability within those SEGs, and the allocation of SEGs for ground/roof support, shotfirer, long hole drill and blast and production and services may require revision. Most of these SEGs show mean exposure values up to double the proposed WEL. However, if measurements were taken as REC, they may have potentially fallen below the proposed WEL.

## 5. Discussion

This study aimed to determine to what extent historical DPM exposure data from the WA mining sector indicate compliance with the current WES and to provide an estimate of potential future non-compliance with the proposed REC-based WEL. Furthermore, the data provided an insight into temporal trends over time and variations attributed to different operational contexts (e.g., underground vs. surface mining) and how these may impact future compliance.

The results indicate that sample means for DPM have reduced significantly over the last decade, presumably due to improved controls and the use of low-sulphur diesel [5], with 91.4% of samples submitted over this period compliant with the current WES of 0.1 mg/m^3^. A further 78.3% of the samples were less than 50% of the exposure standard, and 70.9% were less than 35% of the exposure standard. However, fewer than 30% of samples would comply with the proposed WEL of 0.01 mg/m^3^, without accounting for the fact that historical samples were measured as SMEC, whereas the WEL will be based on REC. The decreasing trend observed in this study aligns with broader trends identified by Rumchev et al. [1], who found a significant reduction in DPM exposure concentrations among WA miners between 2006 and 2012, likely attributable to technological advancements and stricter regulatory practices. The significant downward trend in the data is encouraging, and this continued beyond 2012. The WA mining industry has been largely compliant since 2014 with the current WES 0.1 mg/m^3^ EC. However, as the change per year reduces over time, it is likely that further reductions will be challenging to achieve. Notably, Figure 1 shows a slight increase in mean DPM concentrations from 2023 to 2024, suggesting that further reductions may be increasingly difficult to achieve.

The proposed WEL of 0.01 mg/m^3^ is based on REC, whereas the current WES is based on SMEC. While both are intended to capture DPM, the sampling fractions differ and are not directly comparable. This discrepancy presents a challenge for transitioning between standards and for interpreting historical exposure trends. Attempts to estimate respirable concentrations from historical SMEC measurements using general conversion ratios carry significant uncertainty and are not suitable for compliance assessments. There are studies that provide insights into the proportion of respirable dust particles within the inhalable dust fraction in mining environments, including both surface and underground settings. One such study by Scheepers et al. [19] reported that the geometric mean ratio of respirable to inhalable dust levels was approximately 0.5:1 across European mining sites, with higher ratios observed in underground environments compared to surface operations. Direct comparison should be avoided unless supported by particle size distribution data or co-located sampling studies. Due to the size and complexity of the DEMIRS database, such historical adjustments are not likely to be reliable and would not provide much value to the industry or the regulator. It would, however, be viable to conduct further studies after the introduction of the new sampling regime in 2026 to determine how results trends have changed over time, and there is also an opportunity to conduct future studies to establish what the relationships between the two methods are in the WA mining sector.

Over the last decade, diesel particulate filters and higher-tier engines (such as Euro VI and Tiers IV and V) have become the norm rather than the exception on WA mine sites, including the use of ultra-low-sulphur fuel and low-ash lubricating oils. These improvements have clearly had a positive impact on the quality of exhaust emissions, specifically the DPM component [5]. Mensah et al. [11] also highlight the significant impact of improved diesel engine technologies and emissions control practices, including better engine maintenance and the implementation of exhaust after-treatment technologies, in substantially lowering DPM exposure in underground mining operations. In addition to these improvements, there has been a transition to alternative fuel forms, particularly in underground environments.

Based on the most recent data, the industry sector requiring further improvement is underground mining, and in WA, this primarily involves gold mining operations. Specific SEGs to be managed include ground/roof support, shotfirers, long hole drill and blast, and production and services. These workers typically are located outside of vehicles in close proximity to diesel exhaust. These SEGs may require re-classification to address the variability in the GSD observed; however, this will be difficult to standardise across the sector, given the size and scope of sampling activities across the state. Mensah et al. [11] similarly identified that operators involved in shotcrete application, drilling, and blasting typically experience significantly higher DPM exposures, underscoring the need for targeted control measures in these high-risk occupational groups.

In Australia, the protection provided by respiratory protective equipment (RPE), with adequate fit testing and RPE management procedures in place, can be factored into exposure assessments from December 2026 [20]. This will impact occupational hygiene practice and reporting, and the industry regulators will need to develop ways in which organisations can report their workers’ exposures, considering RPE protection factors as a part of their compliance monitoring. Rumchev et al. [1] stress the necessity of rigorous adherence to respiratory protection usage protocols, as they observed that a high proportion of miners who neglected to wear protective equipment experienced elevated exposures and a higher prevalence of respiratory symptoms.

## 6. Conclusions

This assessment of DPM exposures in the WA mining sector highlights substantial progress in reducing airborne concentrations over the past decade, due to the introduction of higher-tier diesel engines, improved fuel quality, and enhanced emission controls. However, the proposed WEL of 0.01 mg/m^3^ REC, set to take effect in December 2026, presents a significant compliance challenge, especially for underground mining operations where diesel-powered equipment remains prevalent. Trend modelling indicates that while there has been compliance with the current WES, further reductions will be difficult without strategic changes, including increased use of alternative fuels and battery-electric vehicles.

The study identifies specific SEGs, such as ground/roof support and long hole drill and blast, as priority areas for further risk reduction. It also underscores the need for integrating RPE protection factors into exposure assessments and compliance reporting frameworks. Importantly, the transition from SMEC to REC as the regulatory measurement standard introduces methodological challenges that limit direct comparisons with historical data. Addressing this disconnect will be essential to ensure future compliance monitoring is both accurate and consistent. There are opportunities to do further research to determine the relationship between SEMC and REC sample results by running both sampling methods side by side. Once such relationships are determined, retrospective data could be adjusted accordingly.

Moving forward, collaboration between industry, regulators, and occupational hygienists is critical to developing practical and enforceable strategies. Compliance with WHS regulations requires prevention of overexposure above the DPM WES so far as is reasonably practicable.

## Figures and Tables

**Figure 1 ijerph-22-01412-f001:**
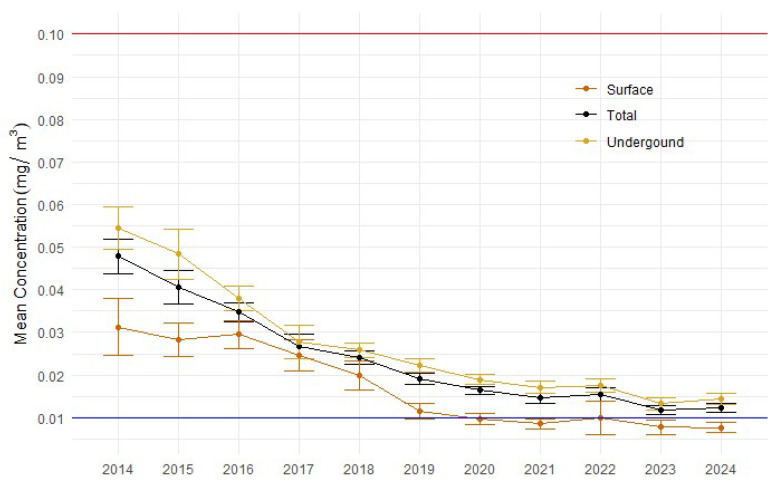
Mean DPM exposures (mg/m^3^) from 2014 to 2024, including LCLs and UCLs, compared against the current WES (red line, 0.1 mg/m^3^) and proposed WEL (blue line, 0.01 mg/m^3^).

**Figure 2 ijerph-22-01412-f002:**
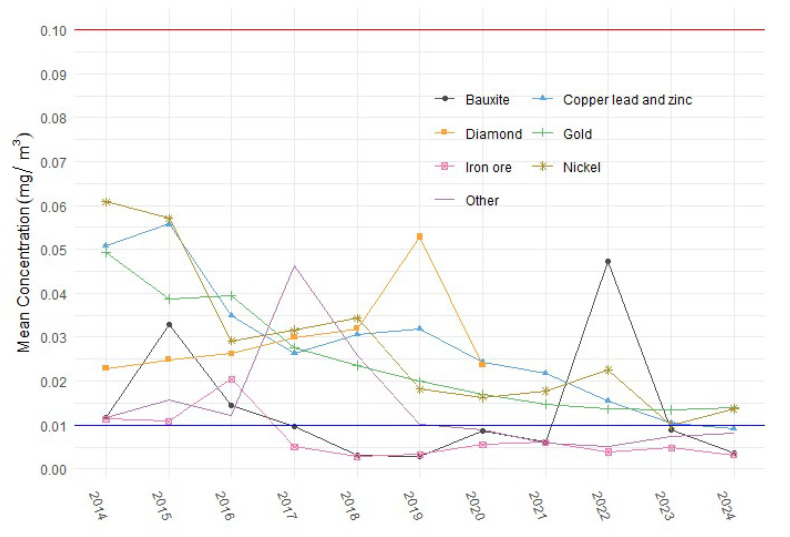
Mean DPM exposure concentrations by commodity from 2014 to 2024 compared against the current WES (red line, 0.1 mg/m^3^) and proposed WEL (blue line, 0.01 mg/m^3^).

**Figure 3 ijerph-22-01412-f003:**
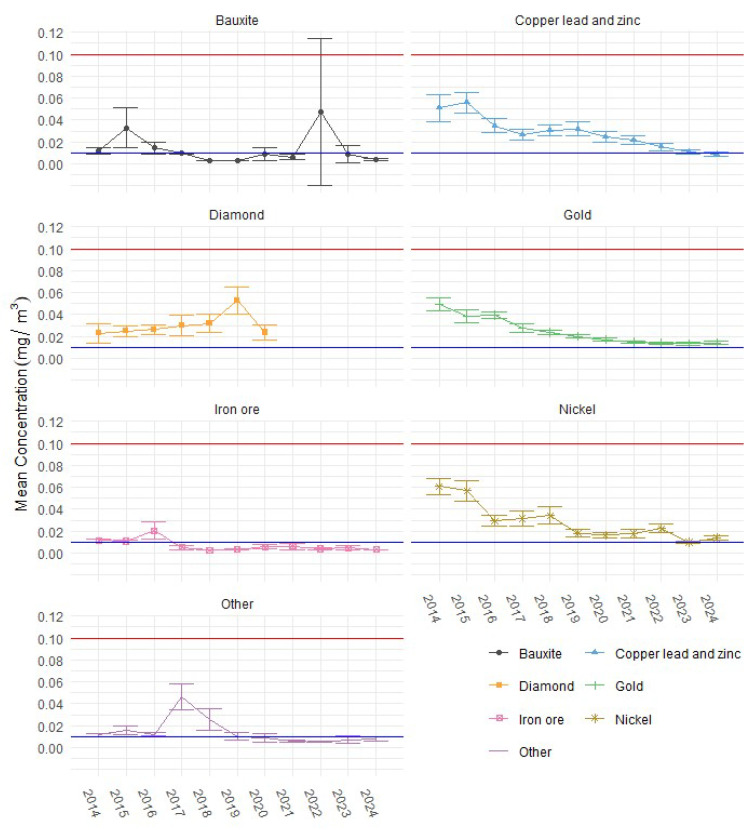
Mean DPM exposure concentrations by commodity from 2014 to 2024, including 95% LCLs and UCLs, compared against the current WES (red line, 0.1 mg/m^3^) and proposed WEL (blue line, 0.01 mg/m^3^).

**Figure 4 ijerph-22-01412-f004:**
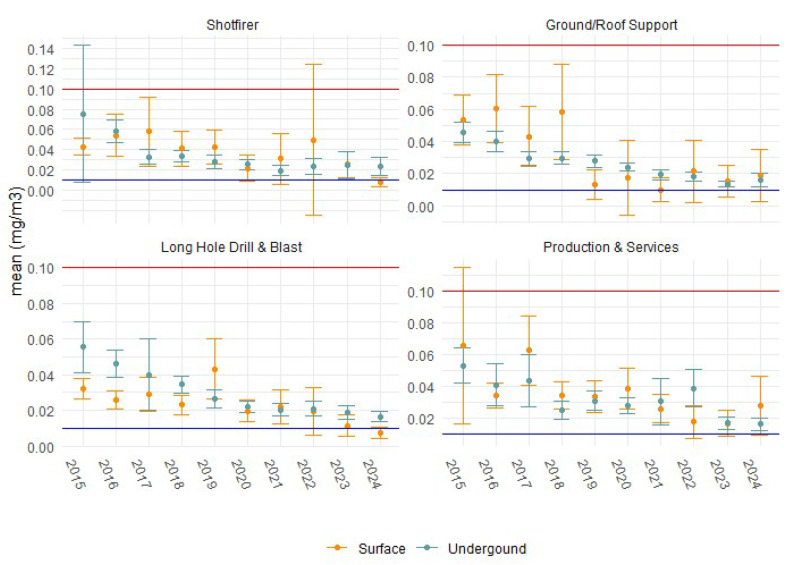
Mean DPM exposure concentrations and 95% LCLs and UCLs for selected occupations from 2015 to 2024 compared against the current WES (red line, 0.1 mg/m^3^) and proposed WEL (blue line, 0.01 mg/m^3^).

**Table 1 ijerph-22-01412-t001:** Number (n) of DPM samples, stratified by commodity and location.

Commodity	Surface (*n*)	Underground (n)	Total (*n*)
	174	0	174
Chemical	20	0	20
Coal	64	0	64
Copper, lead, and zinc	313	1799	2112
Diamond	56	654	710
Gold	2929	11,269	14,198
Iron ore	1644	8	1652
Mineral sands	93	0	93
Nickel	716	3710	4426
Not specified	46	0	46
Other	752	1	753
Rare earth	97	0	97
Silica	53	0	53
Tin, tantalum, and lithium	57	4	61
Total	7014	17,445	24,459

**Table 2 ijerph-22-01412-t002:** DPM sampling results for 2023 and 2024 for selected occupations deemed to have the highest exposures in underground mining.

	Occupation	Total (*n*)	Mean (mg/m^3^)	Min (mg/m^3^)	Max (mg/m^3^)	SD	LCL	UCL	GSD
2023	Shotfirer	62	0.02	<0.001	0.41	0.05	0.012	0.038	2.90
	Ground/Roof Support	231	0.01	<0.001	0.11	0.01	0.012	0.016	2.52
	Long Hole Drill and Blast	171	0.02	<0.001	0.20	0.03	0.015	0.022	2.89
	Production and Services	91	0.02	<0.001	0.13	0.02	0.013	0.021	2.78
2024	Shotfirer	62	0.02	<0.001	0.18	0.04	0.015	0.033	3.25
	Ground/Roof Support	185	0.02	<0.001	0.31	0.03	0.012	0.020	2.71
	Long Hole Drill and Blast	144	0.02	<0.001	0.11	0.02	0.014	0.019	2.62
	Production and Services	131	0.02	<0.001	0.17	0.02	0.012	0.020	2.84

## Data Availability

All data supporting the findings of this study are contained within the confidential DEMIRS database of the Western Australian Government. The extracted de-identified data can be requested from the corresponding author.

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
