# Peer review of "Occupational Exposure to Diesel Particulate Matter in Western Australian Mining: A Retrospective Analysis and Challenges to Future Compliance"

_ijerph, 2025, doi:10.3390/ijerph22091412_

Round 1
Reviewer 1 Report
Comments and Suggestions for Authors
While diesel particulate matter (DPM) exposure levels have significantly declined, compliance with new standards remains challenging. Over the past decade (2014–2024), DPM exposure among Western Australian mining workers has dropped substantially, with arithmetic means far below the current Workplace Exposure Standard (WES) of 0.1 mg/m³ (SMEC). However, when measured against the proposed new Workplace Exposure Limit (WEL) of 0.01 mg/m³ (REC), fewer than 30% of historical samples comply, indicating severe challenges in future compliance. Underground miners (particularly in gold mining) exhibit significantly higher mean DPM exposure than surface miners (p<0.0001), primarily due to enclosed spaces and high usage of diesel equipment in underground operations. Mean DPM exposures in gold, nickel, and copper-lead-zinc mines still exceed the proposed WEL (0.01 mg/m³ REC). High-risk Special Exposure Groups (SEGs) include Ground/Roof Support workers, Shotfirers, Long Hole Drill & Blast workers, and Production & Services workers, with exposure levels up to twice the new standard due to close contact with diesel exhaust.
Limitations and Recommendations:
- The study explicitly notes that historical data are based on SMEC (submicron elemental carbon) measurements, while the new WEL relies on REC (respirable elemental carbon). Direct comparison is technically difficult due to differences in measurement methods and captured particle sizes (REC includes particles up to 10 microns, whereas SMEC excludes those >1 micron). The lack of co-located REC measurements or validated conversion factors may lead to inaccurate predictions of future compliance. The uncertainty range of NIOSH 5040 method at 23 μg/m³ is ±16.7%, exacerbating errors in exposure concentration assessment—especially near the limit, where compliance is hard to judge precisely. The absence of effective conversion methods or co-sampled data between SMEC and REC weakens the reference value of historical data for the new standard, which should be explicitly stated in the results.
- The study relies solely on the database of Western Australian mining regulators, potentially failing to reflect exposure scenarios in other regions or countries, thus limiting the generalisability of findings.
- When analysing trends using a Zero-Inflated Gamma (ZAG) model, the paper does not sufficiently discuss the model’s sensitivity to extreme values or zeroes, which requires further discussion.
- For certain SEGs (e.g., Shotfirers), the Geometric Standard Deviation (GSD) approaches or exceeds 3, indicating large within-group exposure variations. The paper does not propose specific solutions (e.g., reclassifying SEGs or increasing sample size).
- While recommending reduced use of underground diesel equipment, the paper does not evaluate the cost, maintenance, or performance limitations of battery or LPG equipment in practical operations, potentially underestimating industry transition difficulties. A comparison of diesel, battery, and LPG equipment in these aspects is needed.
- Although the new standard is mentioned to align with international norms, the paper does not analyse compliance experiences or lessons from other countries (e.g., US MSHA or EU standards).
- The conclusion comparing confidence intervals with the WEL in Figure 3 requires more detailed explanation to avoid reader misinterpretation.
- The data comparability issue is mentioned multiple times in "Limitations" and "Discussion"; integrating these discussions would enhance logical coherence.
Author Response
See attached response and also the edited manuscript in track changes.

Reviewer 2 Report
Comments and Suggestions for Authors
The manuscript addresses an important topic, but several aspects require clarification and improvement.
Abstract
The abstract does not describe the methods used to analyze the 24,459 records.
The term “enhanced control strategies” is not defined. It should be illustrated with technical measures such as specific technologies, maintenance procedures, or substitution policies.
It is not explained how the analysis of historical data can estimate compliance with the new REC-based WEL, given the measurement differences from SMEC.
Section 1. Introduction
Provides context on DPM sources, health risks, and regulatory changes.
The statement that SMEC data may indicate REC compliance needs stronger qualification.
Uses vague terms such as “targeted interventions” and “control strategies” without concrete mining-related examples.
Does not clearly define the research gap or the practical relevance. Needs a clearer statement of the problem, a defined objective, and a short note on the study’s contribution.
Section 2. Materials and Methods
Describes data extraction, storage, and ethics approval, but does not explain why the chosen dataset is relevant for addressing the research aim.
Statistical methods are listed, but the rationale for selecting these approaches is not provided.
The description of the zero-adjusted gamma model lacks detail on why it is appropriate for the dataset and research question.
The section acknowledges the methodological difference between SMEC and REC measurements but does not explain how this limitation was managed in the analysis beyond noting the problem.
No information on data cleaning steps, exclusion criteria, or potential biases in the dataset.
- Section Limitations
The section lists relevant methodological constraints but does not quantify their potential impact on the results. Authors could indicate how variability in sampling conditions and reporting accuracy may affect exposure estimates.
The limitations related to the absence of validated SMEC–REC conversion factors are acknowledged, but the implications for interpreting compliance with the proposed WEL could be explained more directly.
The statement that SMEC values are expected to overestimate REC is not supported by data from this study.
Section 4. Results
The results section does not mention the SMEC–REC conversion, which is important for interpreting compliance with the new WEL.
The phrase “may have potentially fallen below the proposed WEL” is vague and not supported by data in this section.
Section 5. Discussion
The discussion section repeats numerical findings from the Results section without additional interpretation or analysis. The section should be revised to remove descriptive repetition and focus exclusively on the interpretation of the findings.
The SMEC–REC comparability issue is mentioned, but without concrete approaches for addressing it in future work.
Control measures such as engine technology and alternative fuels are listed but not linked to measured impacts in this study.
Recommendations for SEG reclassification and targeted controls are given without details on feasibility or implementation.
The manuscript does not state the study’s purpose, practical relevance, or scientific contribution. Authors should clarify the rationale behind conducting this research and explain how it contributes to existing knowledge, including identifying the risks occupational groups are exposed to, the effects of these risks, and the measures proposed to address them.
The discussion mentions the study’s limitations but does not suggest how future work could address them.
Section 6. Implications for Practice
Some points repeat information already presented in the Results section (e.g., bullets 1, 3, 4).
The text does not explain how these implications can be applied in practice by the industry but only lists them.
A connection between the study’s findings and these recommendations would be useful, especially regarding the SMEC–REC transition and strategies for reducing underground exposures.
The mention of RPE and the protection factor should be supported with concrete examples of implementation and potential challenges.
Section 7. Conclusions
Repeats information from Results and Implications for Practice rather than providing a final synthesis, with limited synthesis or integration of findings into the broader occupational health context.
Does not address how study limitations influence the interpretation of the conclusions.
Mentions the shift from SMEC to REC without estimating the expected change in measured concentrations.
No recommendations for future research directions, despite methodological challenges being acknowledged.
Comments on the Quality of English LanguageI prefer not to comment on the language quality and recommend that it be reviewed by someone with expertise.
Author Response
See attached as well as the revised manuscript in track changes.

Round 2
Reviewer 1 Report
Comments and Suggestions for Authors
accept
Author Response
Thank you for your review and assessment.
Reviewer 2 Report
Comments and Suggestions for Authors
While the manuscript has been improved following the first-round comments, several issues remain insufficiently addressed and require further revision.
Section 1. Introduction
The introduction has been improved with a research objective and a more qualified discussion of the SMEC–REC comparability issue. However, the research gap and the practical relevance of the study are only implied and should be stated more explicitly to frame the contribution of the work
- Section Limitations
The limitations section acknowledges the SMEC–REC comparability issue. The statement that SMEC values overestimate REC is conceptually correct, but is not supported by data from this study. It should be presented as an assumption based on published literature rather than as a finding.
Section 4. Results
The results present exposure estimates against both SMEC- and REC-based limits, but the methodological limitation of comparing SMEC data to the REC-based WEL is not acknowledged in this section. In addition, the phrase “may have potentially fallen below the proposed WEL” remains and is not supported by data in this section.
Equation 1 must be cited. It must be specified why it was used and where the results of its application were employed.
Section 5. Discussion
The discussion now states the study aim and expands on the SMEC–REC issue, but it still repeats numerical results instead of focusing on interpretation. Control measures and SEG reclassification are mentioned without direct connection to the study’s data or implementation details. The practical relevance and contribution are not stated, and the section does not indicate how the limitations could be addressed in future work
At line 253, avoid the word “presumably”. Present the technical-administrative measures that were applied and led to the reduction of WES, as well as whether they can be applied in cases of REC exceedances.
Section 6. Conclusions
The Conclusions section highlights the SMEC–REC transition and identifies priority SEGs, but it still repeats descriptive results. At line 335, avoid the word “presumably”. The study’s contribution and relevance are not explicitly stated, the influence of methodological limitations on the conclusions is not discussed, and no directions for future research are provided
Author Response
Thank you for your review, please see attached.
